# Effect of Pork Skin Gelatin on the Physical Properties of Pork Myofibrillar Protein Gel and Restructured Ham with Microbial Transglutaminase

**DOI:** 10.3390/gels8120822

**Published:** 2022-12-12

**Authors:** Chang Hoon Lee, Koo Bok Chin

**Affiliations:** Department of Animal Science, Chonnam National University, Gwangju 61186, Republic of Korea

**Keywords:** gelation, rheology, meat protein, meat product

## Abstract

The goal of this study was to determine the qualities of pork myofibrillar protein (MP) gels added with pork gelatin and transglutaminase (TGase), as well as their application to restructured ham (RH). MP mixtures were prepared with various levels of gelatin (0.5, 1.0, and 1.5%. *w*/*w*) and TGase. In this study, cooking loss (CL), gel strength, shear stress, and the microstructure of MP with various levels of gelatin were evaluated. After RHs were manufactured with varying levels of gelatin and TGase, the physicochemical and textural properties were measured. The CL of the MP with 1% (*w*/*w*) of gelatin was decreased. Regardless of the presence of TGase, increased amounts of gelatin in the MP gels resulted in high shear stress. Shear values were higher in the RH with gelatin treatment than in the other treatments. In addition, the RHs with gelatin alone or combined with TGase had high water-holding capacity. The RH with the combination of gelatin and TGase had higher sensory attributes than the control. Gelatin improved the physical properties of the RHs and is recommended for application in various meat products.

## 1. Introduction

Gelatin is extracted from the collagen of animals such as pigs, cows, and fish by acidic and alkaline treatments. It has been generally applied to various foods, beverages, and pharmaceuticals. It is utilized in meat products as a gelling and binding agent because it has a high number of polar amino acids, which improves the hydration qualities [1,2].

Transglutaminase (TGase) is an enzyme that connects glutamic acid and lysine by forming an ε-(γ-glutamyl)lysine bond [3]. The intact muscles were assembled by activating enzymes in the meat proteins when restructured products were treated with TGase, resulting in better textural properties and stability [4]. According to Santhi et al. [5], TGase-catalyzed meat products enhanced gelation, cooking loss, water-binding ability, and emulsion stability. However, excessive cross-linking caused a decrease in gel network development by disrupting the intermolecular aggregation [6]. Furthermore, by interfering with hydrophobic contact between the protein chains, excessive TGase addition reduced the functional qualities such as gel formation and water-holding capacity [7].

Restructured meat products have been produced to expand the sorts of low-value meat accessible in the market. Low-value pork cuts were used to create restructured ham (RH) using tumbling and massaging processes. Ramírez et al. [8] used these techniques to extract salt-soluble proteins to bond the meat pieces. Adding 0.1% (*w*/*w*) TGase to RHs increased their ability to bond without the need for salt or heat [9]. According to Ramírez et al. [8], adding 0.3% (*w*/*w*) TGase to RH increased the textural qualities. The goal of this study was to compare the physicochemical characteristics of myofibrillar protein (MP) with TGase-catalyzed pork skin gelatin. The physical and structural changes of restructured products with varied quantities of gelatin with or without TGase were determined using the model study.

## 2. Results and Discussion

### 2.1. Physical Properties of Myofibrillar Protein Gels

Table 1 shows the CL and gel strength (GS) values of the MP gels with or without TGase and different amounts of gelatin. The results were pooled together by TGase addition and gelatin level because no interactions between TGase and gelatin levels were observed for these particular parameters. The CL of the MP gels with TGase increased when the MP gels with TGase were compared with the MP gels without TGase (*p* < 0.05). This was partly due to lysine–glutamine cross-linking, which was primarily found during incubation [9], resulting in the release of water molecules from the internal structure of meat protein and an increase in GS. However, adding gelatin reduced the CL, and increasing the amount of gelatin reduced the CL correspondingly (*p* < 0.05). Gelatin chains could be covalently cross-linked to form matrices, resulting in a decrease in the CL [10]. According to Feng et al. [11], gelatin increased the weight gain ratio in fish balls due to its high water-holding capacity and hydrophobicity. Moreover, gelatin has a great attraction for water molecules due to the dipole interactions [12]. Gelatin micelles prevented water molecules from preserving the juiciness of the chicken ball [13].

Adding TGase to the MP gels increased the GS due to the formation of the cross-links between meat and non-meat proteins (Table 1). During incubation, enhanced cross-linking between lysine and glutamine may have elevated the GS. Han et al. [7] reported that the presence of TGase enhanced the gel matrix formation. However, there was no difference in GS across the varied amounts of gelatin in the present study (*p* > 0.05). Norziah et al. [14] noted that the GS of gelatin differed according to the protein composition, amino acid concentration, and extraction process. Furthermore, the presence of high proline and hydroxyproline content in gelatin may contribute to its greater GS because the hydroxyl group of hydroxyproline binds with free water molecules to form a stable triple-helix structure [15]. On the other hand, GS was not affected by increasing gelatin levels. High amounts of gelatin produced a strong gel due to the multiple junction zones in the gelation [16]. Brewer et al. [17] observed that combining MP and gelatin created weak gels due to the incompatibility of the two proteins. According to Kaewudom et al. [18], gelatin disrupted the development of the MP structure, thereby reducing the textural features of surimi gels.

The addition of gelatin to the MP mixtures affected the rheological properties of the gel systems. As shown in Figure 1, the shear stress was increased with gelatin addition to the MP mixtures. The fibrous tertiary protein structure of gelatin forms the quaternary structure with triple-helix links in MP mixtures and it can give high flexibility and elasticity properties [19]. MP mixtures with gelatin mediated by TGase mixtures have a higher viscosity than those without TGase since TGase can enhance the hardness and elasticity of meat products through linkage among amino acids [3]. Kuraishi et al. [4] also reported that sausages with TGase need higher breaking force due to the strong linking between amino acids in the meat systems. Myosin cross-linking by TGase created a highly flexible gel network with alternation in intra- and intermolecular interactions [20]. Even though TGase had barely cross-linked gelatin in the MP mixture, the MP gels were polymerized via TGase, increasing the viscosity.

Figure 2 shows the protein molecular weight bands of MP with different amounts of gelatin and TGase. In general, the addition of TGase to proteins results in the formation the high-molecular-weight biopolymers. Zhang et al. [21] reported that the aggregated protein was observed by the intermolecular disulfide linkage on the MP mixtures mediated with TGase. However, the myosin heavy chain (MHC) intensity was significantly reduced after adding TGase. According to the study by Li et al. [22], MHC bands were defoamed by adding TGase to the MP mixtures. However, gelatin addition was not effective in the other bands. Since gelatin is a high-molecular-weight protein, the distinct protein band of pork skin gelatin was shown at 100 kDa and 220 kDa [23]. The MP with TGase and gelatin band showed a polymer band at the top of the SDS-PAGE gel.

The three-dimensional structures of the MP gels were observed to determine the interaction between the meat proteins and gelatin (Figure 3). While the MP gels have a globular and dry surface on their structure, gelatin makes the structure flat and moist by increasing the additional levels of gelatin. Especially, the MP gel with 1% (*w*/*w*) gelatin showed more homogenous structures than the other MP gels with other levels of gelatin. According to the study by Sow and Yang [24], hydrophobic interactions among gelatins increased as the addition of gelatin increased, thereby resulting in aggregation to induce a dense structure. However, excessive gelatin addition to the MP gels remained on their surface because gelatin was eluted during heating, which might reduce the gelation characteristics of the gels. Furthermore, the MP gels with TGase showed a homogenous structure due to intermolecular cross-linking among the MPs [25].

### 2.2. Physical Properties of Restructured Ham

The pH and color values of the RHs are similar across the treatments (Table 2). The addition of 1% gelatin or 3% TGase was not effective on the pH and color values. The range of pH values from 5.96 to 6.07 is adequate to manufacture RH without any defects. The RH with gelatin reduced the CL (Figure 4). Lee and Chin [1] reported that gelatin trapped free water in the protein structure in regular-fat sausages inducing the CL reduction. However, the CL did not decrease when gelatin and TGase were added together. There was no correlation between the reduction in the CL and the addition of TGase. Since the RH product is processed using intact muscle, it is hard to strongly bond intra- or inter-protein complexes [4].

The EM and Allo–Kramer shear values of TGase with gelatin and TGase are shown in Figure 5a,b. The addition of gelatin or TGase and the combination of gelatin and TGase together decreased the EM compared with the control. However, there are no differences between only gelatin or TGase addition and the combination (*p* > 0.05). TGase forms a stable gel structure that is uniformly porous whereby trapping the water molecules within the protein structure and enhancing the water-holding capacity and textural properties [26]. Kaewudom et al. [18] reported that surimi made by fish gelatin and TGase decreased EM-inducing water bound by the hydrogen bonds in the fish gelatin. Although gelatin or TGase addition increased the Allo–Kramer shear value, the RH with gelatin and TGase showed low Allo–Kramer shear strength. Ramírez et al. [8] found that RH with 3% (*w*/*w*) TGase and 1% (*w*/*w*) NaCl showed a homogeneous structure and increased hardness values. Hafidz et al. [23] presented that pork skin gelatin contained high concentrations of glycine and proline, which increased binding capacity due to cross-linking between two amino acid molecules. The RH with gelatin alone resulted in higher Allo–Kramer shear values, whereas gelatin with TGase resulted in low shear values that were equivalent to the control (*p* < 0.05). Similarly, Jongjareonrak et al. [6] found that the structure of the gelatin gel was suppressed by intermolecular aggregation by the covalent bonds as TGase was added.

The sensory attributes of the RH with gelatin and TGase are shown in Table 3. The RH with gelatin and TGase had the highest scores on texture, taste, and overall acceptance (*p* < 0.05). Moreover, the RH with gelatin showed the second-highest score for sensory properties. According to the study by Vergauwen et al. [2], gelatin can improve sensory properties because it has a strong water-binding ability due to a large number of polar amino acids and does not have a characteristic odor.

The FTIR of the RH with gelatin and TGase is shown in Figure 6. The area of amide I was quantified using FTIR, showing a quantitative analysis of secondary structures. Analyzing amide I indicated the degree of protein aggregation. The α-helix and unordered structures were observed at 1650 cm^−1^, and β-sheet structures appeared in the regions around 1615–1630 cm^−1^ and 1680–1700 cm^−1^ [27]. Adding gelatin to the MP gels elevated the percent transmittance value in all configurations, as shown in Figure 4, because of the secondary structure of gelatin. Hafidz et al. [23] reported that pork skin gelatin is a high-molecular-weight protein that includes α and β chains. Furthermore, the addition of gelatin and TGase together had the lowest FTIR values when compared with the other treatments, indicating that the RH containing both gelatin and TGase was more aggregated than the other treatments. The TGase-induced cross-linking of MP might alter the structure of the myosin heavy chain, implying that the α-helix structures were decreased [28]. These structural changes led to the creation of strong gels with an ordered structure, which improved the textural qualities [29].

The protein surface hydrophobicity of the RH with gelatin or in combination with TGase is depicted in Figure 7. The RH with gelatin produced the most BPB bonds and the highest hydrophobicity compared with the other treatments (*p* < 0.05). Multiple hydrophobicity sites are found in meat proteins, and gelatin has many hydrophobicity areas on its surface [11]. Amphiphilic behavior, such as foaming and emulsifying capabilities, is influenced by the ratio of hydrophobic to hydrophilic amino acids [2]. The aggregation of meat proteins was aided by high hydrophobic forces among proteins, while the hydrophobicity of the RH with the addition of gelatin and TGase was reduced [30]. The hydrophobicity domains of gelatin may have been buried inside the meat proteins when the TGase interacted with meat protein and gelatin. In this study, the combination of gelatin and TGase reduced protein surface hydrophobicity. The heating procedure may reveal the interior hydrophobic amino acids in meat acid.

The disulfide bonds were used to assess the sulfhydryl group concentration of meat proteins. The quantitative measurement of peptides by the sulfhydryl groups of the RH with gelatin and TGase is shown in Figure 7. The RH with gelatin had the same amount of sulfhydryl as the control (*p* > 0.05). Sun and Holley [20] reported that gelatin lacking cysteine and cystine was not bound by disulfide, thereby resulting in shorter polypeptide chain lengths during gel formation. Other chemical ingredients are required to expand the length of polypeptide chains. On the other hand, the RH with TGase or in combination with gelatin lowered sulfhydryl levels, resulting in the disulfide-bonding aggregation of meat structures. Bulaj [31] found that disulfide bonds stabilized the creation of folded proteins and lowered entropy, thereby improving thermodynamic stability.

## 3. Conclusions

The CL of the MP gels with gelatin of more than 1% was reduced. The shear stress was increased with increasing the gelatin, regardless of adding TGase. SEM results showed moist and flat structures as the amount of gelatin in the MP gel increased. Based on the MP gel experiments, we decided to use 1.0% gelatin in the manufacture of the RHs since it is a relatively economical concentration. The shear values of the RH with gelatin were higher than the other treatments. The RH with TGase or in conjunction with gelatin had a greater water-holding capacity than the control. The RH with gelatin or in combination with TGase was shown to have improved texture, taste, and overall acceptability in sensory testing. Therefore, pork skin gelatin combined with TGase has the potential to improve the physical qualities of RH and is suggested for use in a variety of meat products.

## 4. Materials and Methods

### 4.1. Materials

Pork loin and ham were purchased from a local meat market (Samho Co., Gwangju, Republic of Korea). The excessive connective tissue was discarded, and the lean meat was cubed (approximately 2 cm^3^). Pork gelatin (Gelatin-G) was provided by a private company (Gel-Tech Co., Busan, Republic of Korea). TGase was bought from Ajinomoto Co., Inc. (Activa TG-S, Ajinomoto Co., Inc., Kawasaki, Japan).

### 4.2. Preparation of Myofibrillar Protein Gels with Gelatin and Transglutaminase

Pork loin cuts were put into a solution of 50 mM phosphate and a 0.1 M NaCl buffer and blended using a mixer (Bowl Rest Mixer, Hamilton Beach/Proctor-Silex, Inc., Southern Pines, NC, USA). The mixture was rinsed with the 0.1 M NaCl buffer three times before being filtered through cotton gauze to remove the connective tissue. An MP pellet was obtained by centrifugation at 1590× *g* for 15 min. The buffer solution adjusted the final protein concentration to 4% (*w*/*w*). The MP pellet was mixed with gelatin (0–1.5%, *w*/*w*) or TGase (0% or 0.5%, *w*/*w*) and placed into vial tubes. All vial tubes were heated from 20 to 80 °C at a rate of 3 °C/min after incubation at 4 °C for 4 h.

#### 4.2.1. Cooking Loss and Gel Strength of Myofibrillar Protein Gels

The difference in gel weight before and after cooking was used to measure cooking loss (CL, %). Gel strength (GS) was determined using the Instron Universal Testing Machine (model #3344, Canton, MA, USA). A steel drill chuck (33BA ½-20, Jacobs chuck, Sparks Glencoe, MD, USA) was used to penetrate 12 mm into the gels and recorded the first peak of breaking force at a speed of 500 mm/min.

#### 4.2.2. The Viscosity of Myofibrillar Protein Mixtures

A concentric cylinder-type rotating rheometer (model #RC30, Rheotec Messtechnik GmbH, Ottendorf-Okrilla, Germany) was used to measure shear stress with a shear rate range of 0 to 600/s using a CC14 coaxial cylinder spindle. The rheometer probe container was filled with 3 mL of the unheated MP mixtures.

#### 4.2.3. Sodium Dodecyl Sulfate-Polyacrylamide Gel Electrophoresis of Myofibrillar Protein Mixtures

The polymerization of MP and gelatin catalyzed by TGase was determined using sodium dodecyl sulfate-polyacrylamide gel electrophoresis (SDS-PAGE). This study used a 10% (*v*/*v*) acrylamide separating gel and a 4% (*v*/*v*) stacking gel [32]. After mixing with the sample buffer, the samples were diluted to a 1% (*w*/*v*) protein concentration and placed onto an acrylamide gel. The gel electrophoresis was run at 150 V for 90 min. The molecular weight of the sample was calculated using a standard marker (product #161-0318, Bio-Rad, Hercules, CA, USA).

#### 4.2.4. Microstructure of Myofibrillar Protein Gels

Samples cut into cube shapes (3 mm^3^) were fixed with 2.5% (*v*/*v*) glutaraldehyde solution at 4 °C overnight. The samples were immersed in a 4% (*v*/*v*) osmium tetroxide solution for 5 h. Dehydration was done in low (10%, *v*/*v*) to high (100%, *v*/*v*) concentrations of ethanol after three washes with dilute water. Dried samples were coated with gold using a coater (model #108 Auto Sputter Coater, Cressington Scientific Instruments Ltd., Waterford, UK). A scanning electron microscope (SEM, model #JSM-6610LV microscope, JEOL Ltd., Tokyo, Japan) was used to analyze the microstructure at 1000 times magnification.

### 4.3. Restructured Ham Processing Procedure

Restructured ham (RH) was manufactured following the formulation in Table 4. The brine solution was combined with pork ham cubes for 30 min before being tumbled for an hour in a vacuum tumbler (model#VTS-42, Marblehead, MA, USA). The tumbled mixtures were packed into a fiber casing (90 mm diameter, Nalro-Faser-Huellen, Wiesbaden, Germany), and kept at 4 °C for 12 h. The RH was smoked and heated in a smoke chamber until it reached the internal temperature of 72 °C (Nu-Vu, ES-13, Food System, Menominee, MI, USA). The ambient temperature was controlled so as to not exceed a 5 °C difference from the sample’s internal temperature in order to prevent the sample surface from drying out. The cooked restructured ham was chilled in ice water for 30 min before being kept at 4 °C until utilized.

#### 4.3.1. pH and Color Values of Restructured Ham

The pH and color values were measured on the internal surface of the products. A pH meter (model #MP120, Mettler-Toledo, Schwarzenbach, Switzerland) was used to measure the pH of the samples at five different spots. Color values (CIE L*, a*, and b*) of the samples were examined in six different regions using a color reader (model #CR-10, Minolta, Osaka, Japan).

#### 4.3.2. Cooking Loss and Expressible Moisture of Restructured Ham

The difference in weight before and after cooking was used to calculate CL (%). Heated samples were cut into 1.5 g cubes to measure expressible moisture (EM,%). Three pieces of filter paper (Whatman #3) were wrapped around the cubed samples and placed in 45 mL plastic tubes. The EM was calculated as the difference between the weight before and after centrifugation at 1660× *g* for 15 min [33].

#### 4.3.3. Allo–Kramer Value of Restructured Ham

The Allo–Kramer (AK, kgf/g) value was determined using an Intron Universal Testing Machine (model #3344, Canton, MA, USA). The cooked restructured ham was cut into squares (20 mm × 20 mm × 2 mm) to measure the Allo–Kramer value. The cutting speed was set to 200 mm/min using an AK probe with ten blades.

#### 4.3.4. Fourier-Transform Infrared Spectroscopy (FTIR) of Restructured Ham

Fourier-transform infrared spectroscopy (Frontier-FTIR/NIR Spectrometer, PerkinElmer, MA, USA) was used to quantify secondary structures in the protein backbone. Scanning was conducted at wavelengths ranging from 4000 to 400 cm^−1^. Graphs were created by arranging the wavelength range from 1450 to 1750 cm^−1^, representing the secondary structure of the protein matrix.

#### 4.3.5. Protein Surface Hydrophobicity of Restructured Ham

Protein surface hydrophobicity was determined according to a modification of the method of Chelh et al. [34]. After thoroughly mixing 1 g of sample and 0.5 mL of bromophenol blue (BPB) solution in 15 mL plastic tubes for 10 min, the mixture was centrifuged at 1660× *g* for 15 min. The supernatant was collected and analyzed at 595 nm using a spectrometer (UV-1601, Shimadzu, Kyoto, Japan) with a phosphate buffer blank. The formula for calculating the hydrophobicity index is follows:BPB bound (μg) = 500 μL × (Absorbance_control_ − Absorbance_sample_)/Absorbance_control_

#### 4.3.6. Sulfhydryl Content of Restructured Ham

The sulfhydryl content was determined using a modified Ellman’s method [20] using a 2,2′-dithiobis (5-nitropyridine) DTNP solution. A quantity of 0.1 g of sample was combined with 0.5 mL of DTNP, 1 mL of Tris, and 8.4 mL of distilled water, and a 20 mM phosphate buffer (pH 6.25) was used as a control. A spectrophotometer was used to detect the absorbance at 412 nm after 5 min of incubation at room temperature.

#### 4.3.7. Sensory Evaluation of Restructured Ham

Seven panelists performed the subjective sensory test at Chonnam National University’s Meat Science Laboratory (Gwangju, Republic of Korea). The cooked samples were put on a white plate with three-digit numbers chosen randomly. An 8-point scale was used to evaluate flavor, color, texture, taste, and overall acceptance, with 1 being the most undesirable and 8 being the most accepted.

### 4.4. Statistical Analysis

Each MP experiment was carried out in triplicate and evaluated by a two-way analysis of variance (ANOVA) (2 × 4, TGase addition x gelatin additive levels) using the SPSS 23.0 program (SPSS Inc., Chicago, IL, USA). Each experiment to determine the RH characteristics was performed in triplicate and analyzed by one-way ANOVA. The significance was noted at levels less than 0.05.

## Figures and Tables

**Figure 1 gels-08-00822-f001:**
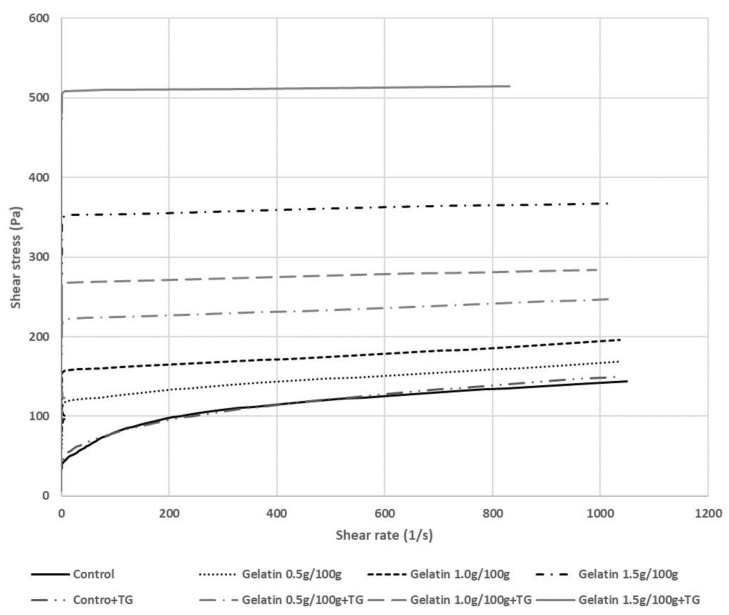
Viscosity of myofibrillar protein mixtures with various additional levels of gelatin and transglutaminase.

**Figure 2 gels-08-00822-f002:**
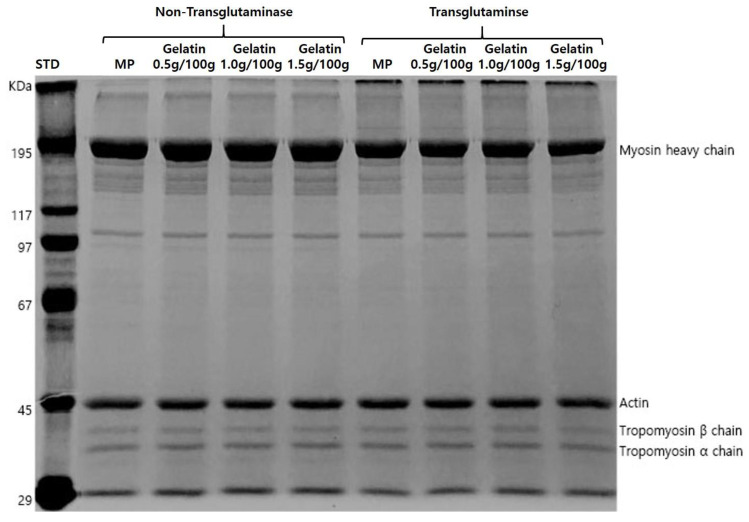
SDS-PAGE of myofibrillar protein mixtures with various additional levels of gelatin and transglutaminase.

**Figure 3 gels-08-00822-f003:**
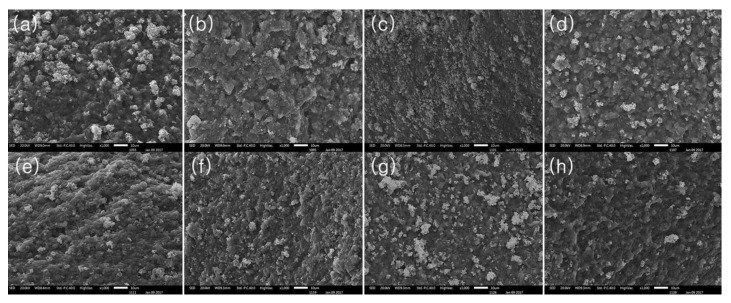
Microstructure of myofibrillar protein mixtures with various additional levels of gelatin and transglutaminase (TGase). (**a**) Control. (**b**) Gelatin 0.5 g/100 g. (**c**) Gelatin 1.0 g/100 g. (**d**) Gelatin 1.5 g/100 g. (**e**) Control with TGase. (**f**) Gelatin 0.5 g/100 g with TGase. (**g**) Gelatin 1.0 g/100 g with TGase. (**h**) Gelatin 1.5 g/100 g with TGase.

**Figure 4 gels-08-00822-f004:**
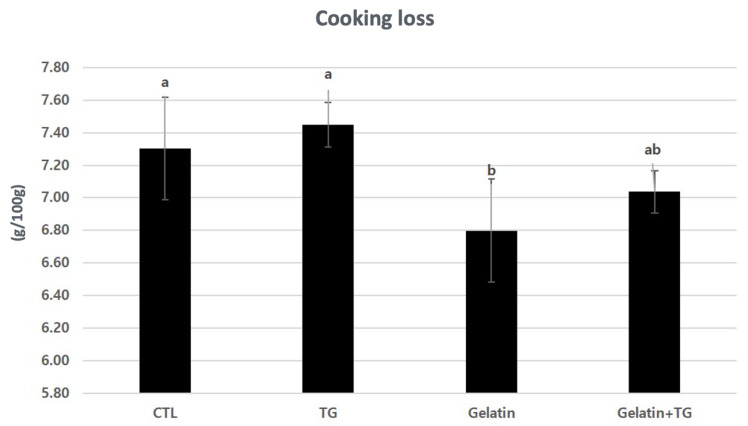
Cooking loss (g/100 g) of restructured ham with gelatin and transglutaminase. ^a,b^ Means (*n* = 3) with the same superscripts in the same row are not different (*p* > 0.05).

**Figure 5 gels-08-00822-f005:**
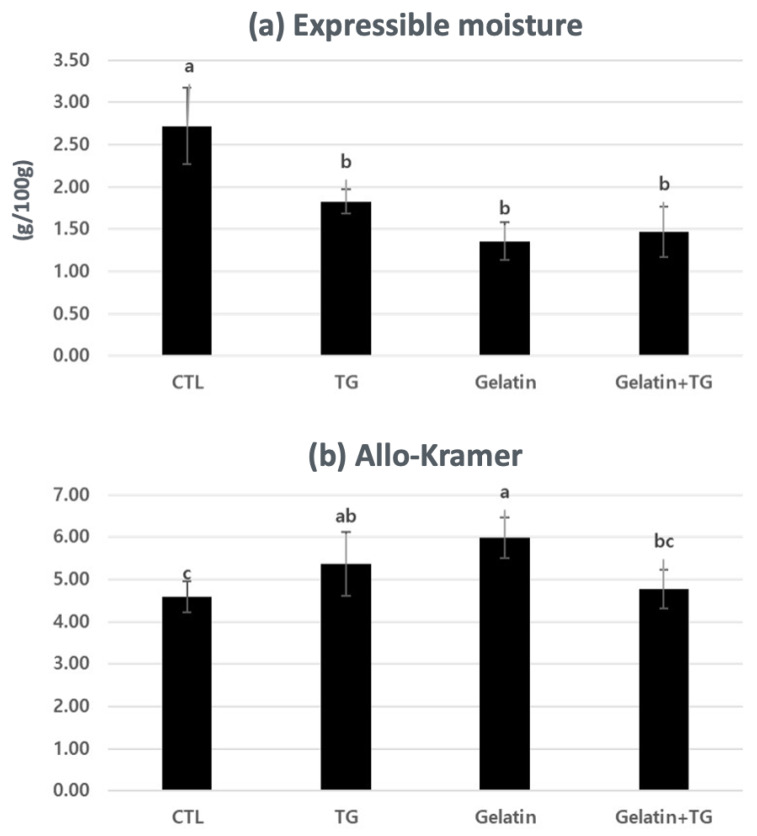
Expressible moisture (**a**) and Allo–Kramer value (**b**) of restructured ham with gelatin and transglutaminase. ^a–c^ Means (*n* = 3) with the same superscripts in the same row are not different (*p* > 0.05).

**Figure 6 gels-08-00822-f006:**
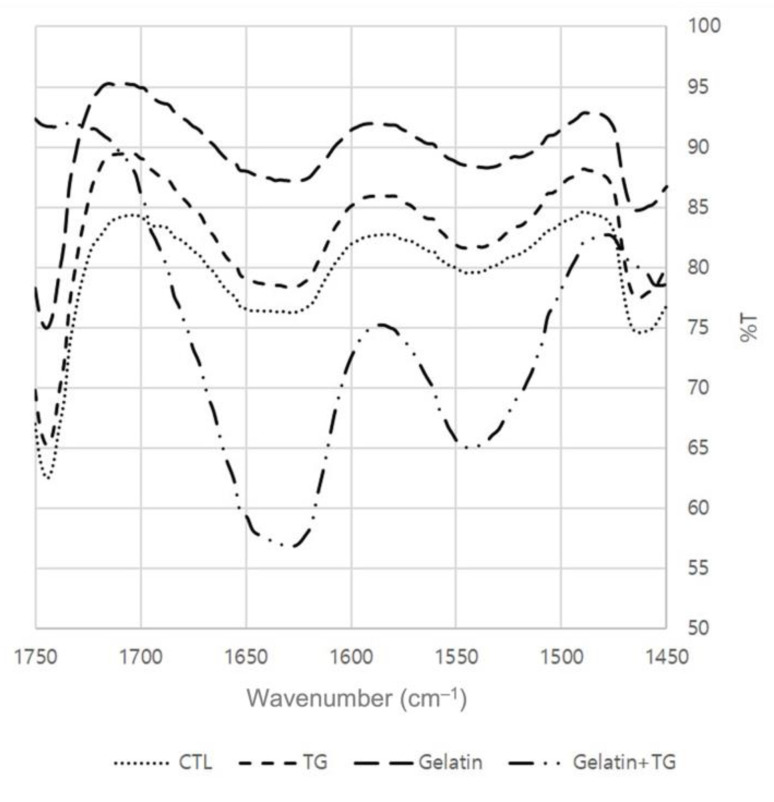
FTIR of restructured ham with gelatin and transglutaminase.

**Figure 7 gels-08-00822-f007:**
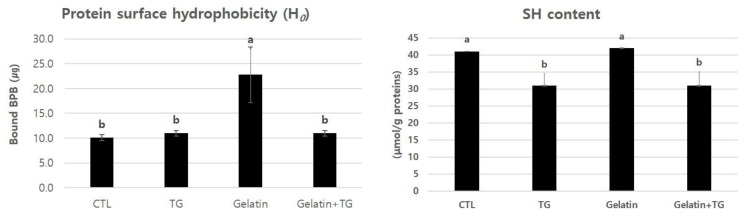
Protein surface hydrophobicity (H_0_) and the contents of peptides by sulfhydryl (-SH) groups on restructured ham with gelatin and transglutaminase. ^a,b^ Means (*n* = 3) with the same superscripts in the same row are not different (*p* > 0.05).

**Table 1 gels-08-00822-t001:** Cooking loss and gel strength of myofibrillar protein gels with various gelatin and transglutaminase additions.

		TGase *		Gelatin Concentrations (g/100 g)			
		NTG	TG	0	0.50	1.00	1.50
Cooking loss (g/100 g)	Mean	9.30 ^b^	13.1 ^a^	18.1 ^a^	12.2 ^b^	8.56 ^b,c^	5.25 ^c^
	S.D.	5.78	6.67	5.42	5.32	3.66	2.40
Gel strength (gf)	Mean	88.7 ^b^	157 ^a^	111	107	130	144
	S.D.	38.9	58.7	60.4	51.9	69.4	65.4

***** TGase: NTG, non-TGase; TG, TGase. ^a–c^ Means (*n* = 3) with the same superscripts in the same row are not different (*p* > 0.05).

**Table 2 gels-08-00822-t002:** pH and color of restructured ham with gelatin and transglutaminase.

	Treatments *				
		CTL	TG	Gelatin	Gelatin + TG
pH	Mean	6.07	5.99	5.98	5.95
	S.D.	0.06	0.11	0.02	0.09
CIE L*	Mean	64.3	64.3	64.3	65.7
	S.D.	0.25	2.10	1.59	0.21
CIE a*	Mean	11.7	11.0	11.2	10.3
	S.D.	0.40	0.73	0.33	1.01
CIE b*	Mean	3.41	3.81	3.58	3.27
	S.D.	0.27	0.28	0.29	0.28

* Treatments: CTL, control; TG, TGase.

**Table 3 gels-08-00822-t003:** Sensory attributes of restructured ham with gelatin and transglutaminase.

	CTL	TGase	Gelatin	Gelatin + TGase
Flavor	5.05 ± 0.46 ^a^	5.48 ± 0.36 ^a^	5.67 ± 0.17 ^a^	5.90 ± 0.22 ^a^
Color	5.24 ± 0.41 ^a^	5.43 ± 0.25 ^a^	5.76 ± 0.22 ^a^	5.62 ± 0.17 ^a^
Texture	4.95 ± 0.46 ^b^	5.33 ± 0.22 ^b^	5.43 ± 0.25 ^a,b^	5.90 ± 0.08 ^a^
Taste	4.76 ± 0.30 ^b^	5.33 ± 0.22 ^b^	5.95 ± 0.36 ^a^	6.24 ± 0.36 ^a^
Overall	4.86 ± 0.52 ^c^	5.57 ± 0.29 ^b^	5.76 ± 0.17 ^a,b^	6.19 ± 0.08 ^a^

^a–c^ Means (*n* = 3) with the same superscripts in the same row are not different (*p* > 0.05).

**Table 4 gels-08-00822-t004:** The formulation of the manufacture of restructured pork ham with pork skin gelatin and transglutaminase.

Ingredients	Treatments * (g/100 g)			
	CTL	TG	Gelatin	TG + Gelatin
1. Meat	83.3	83.3	83.3	83.3
2. Brine solution	16.7	17.0	17.7	18.0
(1) Ice water	12.7	12.7	12.7	12.7
(2) Salt	1.27	1.27	1.27	1.27
(3) Phosphate	0.40	0.40	0.40	0.40
(4) Cureblend **	0.25	0.25	0.25	0.25
(5) Sugar	1.00	1.00	1.00	1.00
(6) Corn syrup solids	1.00	1.00	1.00	1.00
(7) Sodium erythorbate	0.05	0.05	0.05	0.05
(8) Transglutaminase	0.00	0.30	0.00	0.30
(9) Gelatin	0.00	0.00	1.00	1.00
Total	100.0	100.3	101.0	101.3

* Treatments: CTL, control; TG, TGase. ** Cureblend, mixture with sodium chloride (93.75 g/100 g) and sodium nitrite (6.25 g/100 g).

## Data Availability

Not applicable.

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
