# Peer review of "Effect of Pork Skin Gelatin on the Physical Properties of Pork Myofibrillar Protein Gel and Restructured Ham with Microbial Transglutaminase"

_gels, 2022, doi:10.3390/gels8120822_

Round 1

Reviewer 1 Report

Dear authors,

It was an interesting study to read but I still have a some questions and comments. In attachment you can find the manuscript with my comments.

In general it was a good experimental design, but I think there is still some work concerning your text. The introduction is clear and to the point. The ‘Material and methods’ section is still missing a few things as indicated in the manuscript. The methods  can be described more in detail.

In the result section it is  not always clear how your  citations can be linked to your research. Sometimes I miss the coherence in the text. I would also pay attention to the comparison between the results obtained at model level (MP gels) and in the cooked ham. To what extent do they correspond, what are the differences, what is the cause of this,...? I would also mention this in your conclusion.

Author Response

Please see tha attached pdf file

Reviewer 2 Report

General comments:

1- Keywords must be changed because they are mentioned in the title.

2- References citation in the text must be numeral

3-  Titles above each figure must be removed

4- English language must be improved

Other comments:

All other comments are attached as comments in my PDF file

Round 2

Reviewer 1 Report

Dear authors, 

thank you for updating the manuscript. However, there are still a few questions from my first report that are unanswered. I indicate them in the updated manuscript. Could you please formulate an answer to all my questions. Maybe it is better to do this in a cover letter. Also pay attention to the english language. There are still improvements to be made. 

Kind regards!
